# Development of Smart Composites Based on Doped-TiO_2_ Nanoparticles with Visible Light Anticancer Properties

**DOI:** 10.3390/ma12162589

**Published:** 2019-08-14

**Authors:** Evdokia Galata, Eleni A. Georgakopoulou, Maria-Emmanouela Kassalia, Nefeli Papadopoulou-Fermeli, Evangelia A. Pavlatou

**Affiliations:** Laboratory of General Chemistry, School of Chemical Engineering, National Technical University of Athens, Zografou Campus, 9, Iroon Polytechniou str., 15780 Zografou, Greece

**Keywords:** N-doped TiO_2_, Fe-doped TiO_2_, Fe,N-doped TiO_2_, sol gel, IP microgel, pNiPam, drug delivery, photocatalysis, visible light, anticancer

## Abstract

In this study, the synthesis of smart, polymerically embedded titanium dioxide (TiO_2_) nanoparticles aimed to exhibit photo-induced anticancer properties under visible light irradiation is investigated. The TiO_2_ nanoparticles were prepared by utilizing the sol gel method with different dopants, including nitrogen (N-doped), iron (Fe-doped), and nitrogen and iron (Fe,N-doped). The dopants were embedded in an interpenetrating (IP) network microgel synthesized by stimuli responsive poly (N-Isopropylacrylamide-co-polyacrylicacid)–pNipam-co-PAA forming composite particles. All the types of produced particles were characterized by X-ray powder diffraction, micro-Raman, Fourier-transform infrared, X-ray photoelectron, ultra-violet-visible spectroscopy, Field Emission Scanning Electron, Transmission Electron microscopy, and Dynamic Light Scattering techniques. The experimental findings indicate that the doped TiO_2_ nanoparticles were successfully embedded in the microgel. The N-doped TiO_2_ nano-powders and composite particles exhibit the best photocatalytic degradation of the pollutant methylene blue under visible light irradiation. Similarly, the highly malignant MDA-MB-231 breast cancer epithelial cells were susceptible to the inhibition of cell proliferation at visible light, especially in the presence of N-doped powders and composites, compared to the non-metastatic MCF-7 cells, which were not affected.

## 1. Introduction

The scientific community approaches the multivariate condition of cancer disease in various ways. There are still aspects in the research field of alternative cancer treatment that remain to be investigated, which focuses on minimizing the side effects of conventional cancer treatments. Nanomaterials are proven to be a promising candidate for filling this scientific gap, which is a part of their extended range of applications. A wide variety of materials can be used to create nano-particles for drug delivery. The most common materials include proteins, liposomes, polymers, polymer-lipid hybrids, dendrimers, hydrogels, and inorganic materials [1]. For instance, there are recent studies promoting a variety of inorganic nanoparticles (NPs), such as metallic AuNPs [2], AgNPs [3,4], carbon nanotubes [5], SiO_2_ [6], and ZnO [7], as antimicrobial and anticancer agents. In addition, photo-activated TiO_2_ NPs [8,9,10] have been tested and the importance of the exploit of photocatalytic process in anticancer applications has been established. Additionally, chemically modified TiO_2_ nanostructures, such as Ag-TiO_2_ nanofibers [11] and Se-TiO_2_ nanoarrays [12], have been proven to exhibit a significant antitumor performance. It should be noted that a wide diversity of metal oxide photocatalytic materials such as ZnO [13,14], TiO_2_ [15,16], CeO_2_ [17], and WO_3_ [18] has been investigated for both environmental and energy applications. Nevertheless, regarding the anticancer photo-induced treatment, the scientific studies published so far are mainly TiO_2_ oriented, which introduces this metal oxide as a proper applicant for this present research [8,9]. It is strongly advocated that TiO_2_ can cause cyto-toxicity or geno-toxicity and induce apoptosis [19,20,21,22]. It is well established that, when TiO_2_ particles are excited by light, the photon energy generates pairs of electrons and holes that react with water and oxygen to yield reactive oxygen species (ROS) that have been proven to significantly damage cancer cells [23,24]. Therefore, TiO_2_ particles are one of the promising photo-sensitizers against cancer.

From a physical-chemical point of view, TiO_2_ is the most common compound of Ti and occurs in three crystalline polymorphs: rutile (tetragonal), anatase (tetragonal), and brookite (orthorhombic). Rutile and anatase are those with the higher catalytic performance and display higher stability [25,26,27]. TiO_2_ can be photoactivated by ultraviolet (UV) radiation, due to its energy gap (E_g_). However, due to the UV harmful effects in medical applications, which are related to DNA damage (mutations, single strand breaks (SSDs), double strand breaks (DSBs), etc.), it is usually avoided and replaced by other alternatives [28]. To enhance the photocatalytic properties of TiO_2_ under visible light irradiation, doping with non-metal or metal elements is applied [29,30]. The chemically modified TiO_2_ nanoparticles are used in a wide range of environmental applications due to their self-cleaning properties [31,32]. Relatively few studies concerning the anticancer effect of doped TiO_2_ nanoparticles have been reported [33]. In this work, TiO_2_ was chosen to be doped with nitrogen (Ν-doped TiO_2_), iron (Fe-doped ΤiO_2_), and co-doped iron-nitrogen (Fe,N-doped ΤiO_2_), considering their action in leukemia tumors [34] and human epithelial carcinoma cells [35]. Hence, the motivation of this study is the design of a thermo-responsive drug delivery system composed of a stimuli responsive polymer matrix and embedded nano-sized doped-TiO_2_ powders that target a controlled drug release under visible light irradiation.

Stimuli responsive polymers are “smart” polymers that have the ability to respond to slight environmental changes, such as temperature, pH, ionic strength, light, electric field, and magnetic field, depending on their structure [36]. This property makes these polymers preferable candidates for biomedical applications, including the formation of drug delivery systems. Microgels are three-dimensional linked networks that are able to absorb and retain large amounts of water. Poly (N–isoprpopylacrylamide)–pNipam microgel is a typical temperature responsive hydrogel. PNipam is the most studied polymer in drug delivery and biomedical applications due to exhibiting a phase transition at 32–34 °C, which is hydrophilic below this temperature or hydrophobic above it. This lower critical solution temperature (LCST) corresponds to the part in the phase diagram where the entropic gain of the system overcomes the enthalpic contribution associated with hydrogen bonds [37] and is close to the human body temperature [37,38,39]. The selection of the proper co-polymer for the formation of the IP microgel network results in the adjustment of this volume phase transition temperature (VPTT), depending on the demands of the favorable technical application [40]. Given that this study targets the development of a drug delivery system designed to exhibit photo-induced, anticancer properties under visible light, composite TiO_2_-based materials are investigated. To promote the embedding of TiO_2_ nanoparticles, polyacrylic acid-PAA with interpenetrating-IP linear chains has been used as a second monomer [38,41,42]. To the best of our knowledge, the embedding of doped TiO_2_ particles in pNipam IP network microgel has not been reported, while embedded non-doped TiO_2_ particles have been investigated for environmental applications [41,42,43,44]. Recent studies support the idea that TiO_2_ could be used in the photo-thermal therapy of melanoma cancer cells [9], while others advocate that TiO_2_ particles can affect cultured human lymphoblastoid cells [19] or leukemia tumors [34,45]. Therefore, in this study, breast cancer epithelial cells have been chosen in order to investigate the biological effect of doped TiO_2_ particles on cell proliferation, since this is a continuation of our previous studies on breast cancer cells [10,46] simultaneously promoting a novel approach on cancer treatment.

Furthermore, the innovation proposed at this work includes the embedding of doped and co-doped TiO_2_ nanoparticles in an IP network microgel of pNipam-co-PAA, which results in the synthesis of pNipam-co-PAA/(co)doped-TiO_2_ composite particles acting as a thermo-responsive drug delivery system operating in human body environmental conditions. The photo-activation upon visible light irradiation of this system is investigated on breast cancer epithelial cells for the first time, in order to promote a promising photo-sensitizer against cancer. This is an early-stage research focusing on the development of novel materials, designed to be applied as anticancer agents in a biological system in vitro [47].

Summarizing, in the present work, we synthesized chemically modified N-TiO_2_, Fe-TiO_2_, and Fe,N-TiO_2_ nanoparticles by the sol-gel method. An IP network microgel of pNipam-co-PAA was prepared by a precipitation polymerization process, in which the as-produced TiO_2_ nano-powders were embedded and formed pNipam-co-PAA/(co)doped-TiO_2_ composite particles in order to provide a novel thermo-responsive drug delivery system. Full characterization, utilizing techniques like X-ray powder diffraction (XRD), micro-Raman, Fourier-transform infrared (FT-IR), X-ray photoelectron (XPS), ultra-violet-visible (UV-vis) spectroscopy, Field Emission Scanning Electron (FESEM), Transmission Electron (TEM) microscopy, and Dynamic Light Scattering (DLS) was imposed in order to confirm the produced nanoparticles. Composite particles exhibit the desirable physicochemical properties and morphology. The degradation of the methylene blue (MB) pollutant was evaluated for the as-produced nano-powders and composite particles under visible light irradiation. Additionally, cultured MCF-7 and MDA-MB-231 breast cancer epithelial cells were irradiated using visible light in order to investigate the anticancer behavior of the proposed innovative composite materials.

## 2. Materials and Methods

### 2.1. Preparation of Doped TiO_2_ Nanoparticles

For the synthesis of the doped-TiO_2_ powders, three different experimental paths were followed utilizing the sol-gel method.

For synthesizing chemically modified N-TiO_2_ particles, 100 mL of deionized water was acidified with nitric acid (HNO_3_ 65%, Penta, Prague, Czech Republic), in order to adjust the pH being acidic for the evolution of hydrolysis. Additionally, 15 mL of Titanium Butoxide (Titanium (IV) butoxide, C_16_H_36_O_4_Ti, 97%, Sigma-Aldrich, Darmstadt, Germany) alkoxide was then added under vigorous stirring. The resulting solution was whitish. After 5 h, the addition of 30 mL 2-propanol (C_3_H_8_O 99.8%, Sigma-Aldrich, Darmstadt, Germany) followed and the resulting titania sol-gel became transparent. Furthermore, 30 g of urea (CH_4_N_2_O 99%, Sigma-Aldrich, Darmstadt, Germany) were added in the titania sol under vigorous stirring. The solution was heated until the complete evaporation of the solvent was achieved. The gel produced was calcinated at 450 °C for 4 h. The resulting powder was triturated and purified by rinsing and centrifugation to remove impurities and it had a yellow color [10,48,49].

Fe-doped TiO_2_ powder was prepared by hydrolysis through the sol–gel route using Titanium isopropoxide (Sigma-Aldrich, Darmstadt, Germany) and Ferric nitrate (Iron (III) nitrate nonahydrate ≥99.95%, Fe(NO_3_)_3_∙9H_2_O, Alfa Aesar, Ward Hill, Massachusetts, United States) as precursors. In a mixture of 60 vol% ethanol (CH_3_CH_2_OH, anhydrous ≥99.5%, Sigma-Aldrich, Darmstadt, Germany) and water (50 mL), 1.5 mL of titanium precursor compound (Titanium (IV) isopropoxide, C_12_H_28_O_4_Ti, 98%, Acros Organics, Geel, Belgium) was added dropwise under vigorous stirring and then 70 mg Fe(NO_3_)_3_·9H_2_O (Alfa Aesar, Ward Hill, Massachusetts, United States) powder was added. The mixture was hydrolyzed at a boiling temperature for 2 h under vigorous stirring, and then centrifuged at 3000 rpm for 20 min. The resulting gel was dried at 80 °C for 12 h to remove the solvents, and calcinated at 350 °C for 2 h to obtain a sub-yellow powder. Eventually, the powder was ground and used further for characterizations. The calculated atomic ratio (in %) of Fe to Ti in the precursors was 3% [35].

Fe,N-TiO_2_ particles were also prepared by using Titanium Butoxide (0.0556 mol, Sigma-Aldrich, Darmstadt, Germany) mixed with Fe (NO_3_)_3_∙9H_2_O powder (4 × 10^−4^ mol) and dissolved in 60 mL anhydrous ethanol, at room temperature. At the same time, a quantity of hydroxylamine hydrochloride (8 × 10^−3^ mol, NH_2_OH∙HCl, 99%, Alfa Aesar, Ward Hill, Massachusetts, United States) was mixed with 2 mL distilled water and 16 mL anhydrous ethanol. The first solution was added slowly to the second, under vigorous stirring. The final sol-gel was obtained after further stirring for a few minutes. The prepared TiO_2_ gel was rinsed with deionized water and dried at 120 °C for 12 h. Afterward, it was calcinated at 400 °C for 2 h and triturated to retrieve a yellowish-brown powder. According to the synthesis process, a 2 wt% Fe to Ti ratio was prepared [34].

### 2.2. Synthesis of the IP Network Microgel pNiPAM-co-PAA

The IP network microgel pNipam-co-PAA was formed by following a precipitation polymerization process. Monomers of Nipam (1 g, N-Isopropylacrylamide >98%, TCI, Chennai, India) were purified and recrystallized by hexane and dissolved in an aqueous solution (200 mL) together with the cross-linker MBA (4 × 10^−2^ g, N,N-Methylenebisacrylamide 99 + %, Alfa Aesar, Ward Hill, Massachusetts, United States). Then, PAA (1.5 g, MW 15,000 g/mol, polyacrylic acid sodium salt, Sigma-Aldrich, Darmstadt, Germany) was added. The reaction mixture was purged for 1 h with pure N_2_ (99.99999%, Air Liquide Hellas, Athens, Greece) and heated at 75 °C. Afterward, potassium persulfate (2 × 10^−2^ g, Potassium Persulfate >99%. Fisher Chemicals, Hampton, U.S.A.) was added and the polymerization started, which lasted 20 h under vigorous stirring [41,43,44].

To form the embedded composite particles, suspensions of different chemically modified TiO_2_ powders were prepared, using 50 mg of each powder. These were suspended in 20 mL deionized water, after adjusting the pH to 2 using HCl (Sigma-Aldrich, Darmstadt, Germany) to charge them positively, and then added to 20 mL of microgel solution. These composite particles were centrifuged and washed with deionized water three times and dried at 70 °C [43]. The same process was followed for the Evonik P25 powder. The produced composite particles were loaded in 50% titania mass fraction.

### 2.3. Characterization Techniques

In order to study the structural properties of the powders, various techniques were applied such as XRD, micro-Raman, and Infrared Spectroscopy. For the XRD analysis (D8 Advance, Bruker, Germany), the measurements were performed at a 2-theta angle with a range of 20° to 80° and a scanning rate 0.1 °/min employing Cu-Kα radiation (λ = 1.5418 Å) at a voltage of 30 kV and a current of 15 mA. The Raman spectroscopy (inVia, Renishaw, Wotton-under-Edge, Gloucestershire, UK) apparatus used two excitation sources, that of a solid-state laser (λ = 532 nm) and that of a high power near infrared (NIR), diode laser (λ = 785 nm). Raman measurements were performed at room temperature in a backscattering configuration and the laser beam was focused onto the samples by means of an × 50 short distance magnification lens, with low excitation power, so as to secure low laser heating of the samples. The frequency shifts were calibrated by an internal Si reference. Two to three spots were measured for each sample. The exposure time was 10 s, with 2–10 accumulations. For the FT-IR analysis (Spectrum 100, U-ATR with a ZnSe Ri 2.4 crystal, PerkinElmer, Waltham, Massachusetts, USA.) the number of scans performed for obtaining each spectrum was 16, with a resolution of 4 cm^−1^. The band gap of the produced particles was measured using a UV-Vis spectrometer (U-3010, Hitachi, Tokyo, Japan) equipped with a 50 mm integrating sphere, which allows diffuse reflectance measurements. The morphological characteristics of the powders were analyzed by a field emission scanning electron microscope (FESEM, JSM-7401F, JEOL, Tokyo, Japan). The surface chemical states of the samples were measured by X-ray photoelectron spectroscopy (XPS-Particle size analyzer, MasterSizer S, MSS, Malvern Inst., Malvern, UK). The photoemission experiments were carried out in an ultra-high vacuum system (UHV) with base pressure 1 × 10^−9^ mbar using un-monochromatized MgKα line at 1253.6 eV. The XPS core level spectra were analyzed using a fitting routine, which can decompose each spectrum into individual mixed Gaussian-Lorentzian peaks after a Shirley background subtraction. The samples were initially in powder-form and pressed in stainless steel pellets. The size estimation of the P-NiPAM-PAAc microgel was determined via Dynamic light scattering (Zeta Sizer nano S, Malvern Inst., Malvern, UK). Lastly, Transmission Electron Microscopy (TEM, CM20, Philips, Amsterdam, The Netherlands) was used to study the morphology of the composite particles.

### 2.4. Photocatalytic Test

The photocatalytic activity of powders and composites was evaluated under appropriate visible light illumination. Daylight irradiation was provided by four parallel 15 W daylight lamps (350–750 nm, 3 mW cm^−2^, Sylvania, Wilmington, U.S.A.) with a 400 nm cutoff filter, into a lab-made photo-reactor placed at a distance of 10 cm from the samples. Methylene Blue (MB, Sigma-Aldrich, Darmstadt, Germany) was used as the model pollutant for photocatalytic degradation under visible irradiation. The initial concentration of the pollutant was 0.6∙× 10^−5^ M (C_0_ = 0.6). Prior to each photocatalytic run, the solution was saturated in oxygen by bubbling O_2_ gas for 2 h, and, then, 5 mg of doped TiO_2_ powders/hybrids were placed in the glass containers together with 5 mL of the pollutant. The samples were maintained in the dark for 1 h to achieve adsorption-desorption equilibrium. All the photocatalytic experiments were performed under continuous stirring and conducted at three different samples for each type of tested TiO_2_ particles. Methylene Blue solutions display maximum absorbance at 664 nm without a significant shift. Therefore, temporal changes on the concentration of the pollutant were monitored by examining the variations of intensity at the previously mentioned absorption band recorded at the Uv-Vis spectrophotometer (U-2001, Hitachi, Tokyo, Japan). In the photocatalytic test lasting 150 min, the amount of the pollutant was taken every 30 min to measure the concentration of it. The ratio of the measured absorption each time (A) to the original (A_initial_) absorption corresponds to the ratio of *C*/*C*_0_ pollutant concentrations [50].

### 2.5. Biological Effect

#### 2.5.1. Effect on Cell Proliferation

The anticancer effect of the doped TiO_2_ particles and composites was examined by using two cancer cell lines: MDA-MB-231 (human breast adenocarcinoma, highly invasive, ATCC - LGC Standards GmbH, Wesel, Germany) and Michigan Cancer Foundation MCF-7 (low metastatic potential, ATCC - LGC Standards GmbH, Wesel, Germany), which are both derived from breast epithelium. Both MCF-7 and MDA-MB-231 cells were cultured in 75 cm^2^ flasks, in Dulbecco’s modified Eagle’s medium (DMEM) supplemented with 10% fetal bovine serum (FBS), 1% L-glutamine, 1% sodium pyruvate, and antibiotics (all media were purchased from BioWest, Nuaillé, France), and incubated at 37 °C in a 5% CO_2_ incubator. In addition, trypsin–EDTA: 0.05%/0.02% (*w/v*) (Gibco BRL, Life Technologies,Thermo Scientific, Paisley, United Kindom) were used for the trypsinization of cells [10,46].

In order to estimate the effect of the TiO_2_-based particles on the cell proliferation, the cancer breast cells (~100,000 cells/well) were seeded in 6-well plates, using the previously mentioned medium. Twenty-four hours after plating, various concentrations of dispersions of the different types of doped TiO_2_ particles were added to the appropriate plates, and the samples were irradiated using visible light for 2 h. The same photocatalytic reactor that was used in the photocatalytic tests was also used here, avoiding the increase of the temperature by adding a venting system and by suitably setting the distance between the lamps and the samples at 20 cm. The cells were further cultured, and, in the following three days, the cells were stained with Trypan Blue and counted using a hemocytometer (Neubauer, Corning, Amsterdam, The Netherlands) and an Optical Microscope (OLYMPUS IM, Olympus Deutschland GmbH, Hamburg, Germany) [51]. The experiment was repeated several times, in order to determine the minimal time of photo-activation of titanium dioxide dispersions and the optimal concentrations of TiO_2_ particles and composites. The appropriate conditions were selected, and the experiment was repeated at least five times in triplicate. In all instances, similar results were obtained.

Values of the cell number are presented as means ± standard deviation. Statistically significant differences between values were evaluated by one-way analysis of variance and the nonparametric Kruskal–Wallis method in the SPSS program (IBM Corporation, Armonk, NY, USA.). *p* < 0.05 was considered statistically significant.

#### 2.5.2. Oxidative Stress Detection Assay

GSH/GSSG Ratio Detection Assay was employed, in order to quantify reduced and/or oxidized glutathione (GSH) in the tested samples. This kit (Cellular Glutathione Detection Assay Kit, by Cell Signaling Technology, Massachusetts, USA.) uses a non-fluorescent cell permeable dye (monochlorobimane (MCBCell Signaling Technology, Massachusetts, USA.), that becomes strongly fluorescent upon reacting directly with GSH. MCB displays a high affinity for reduced glutathione and exhibits a very low fluorescent yield when free in solution. Upon binding to GSH, the dye exhibits a strong blue fluorescence that can be measured at an excitation wavelength of 380 nm and an emission wavelength of 460 nm (Ex/Em 380/460 nm). For this purpose, the cells (~50,000 cells/well) were seeded in 96-well plates. Twenty-four hours after plating, 0.4 mg/mL of each dispersion of the different types of doped and composite TiO_2_ particles were added to the appropriate plates, and the samples were irradiated using visible light for 2 h. Following treatment, the assay plates were centrifuged for 10 min at 1200 rpm. Digitonin Lysis Buffer was added to each well and incubation on the shaker for 15 min at room temperature was followed. The samples were centrifuged for 10 min, at 14,000 rpm, at 4 °C and proceeded to utilize GSH assay. Thus, the Tris Assay Buffer (Cell Signaling Technology, Massachusetts, USA.) was mixed with the test sample, GSH standard, and working solution in a black 96-well plate with a clear bottom to a final volume of 100 μL. The plate was read at a plate reader [52].

## 3. Results and Discussion

### 3.1. Characterization of the Doped-TiO_2_ Nanopowders

XRD was used in order to investigate the crystallinity of the different chemically modified TiO_2_ powders produced. Figure 1 presents all the diffraction diagrams recorded for the as-produced powders. The dominant crystal phase of TiO_2_ is anatase in all the samples, while the rutile phase is less than 1%. The highest intensity diffraction peak of anatase is at 2θ = 25.35° corresponding to (1 0 1) crystal plain and all the other peaks of anatase spotted are in accordance with the PDF No 03-065-5714. These results are supported by the relevant literature [34,35,53]. Rutile is observed only in the case of N-doped TiO_2_ powder, at 2θ = 27.30° corresponding to (1 1 0) crystal plain. The presence of rutile in N-doped TiO_2_ could be associated with the higher than 400 °C calcination temperature followed in the experimental procedure. It is reported that the transition temperature from pure bulk anatase to rutile in air is in the range of 400–1200 °C [54]. Additionally, it is evident that, when Fe is used as a dopant, the intensity of the diffraction peaks is reduced, accompanied by a broadening of the observed peaks. As reported by Zhou et al. [55], the larger the amount of Fe doping, the wider the width of diffraction peaks and the worse the crystallization of TiO_2._

The average crystallite size of the as-produced powders was calculated by using Scherrer’s equation.
(1)$$d=\frac{0.89\;\lambda }{\beta \cos\theta }$$ where d is the average crystalline size, 0.89 is the Scherrer’s constant, λ is the X-ray wavelength, θ is the diffraction angle, and β is the FWHM (full-width-half-maximum). This is calculated for the main peak of anatase (1 0 1) at 2θ = 25.35° [56]. The average crystallite size of all the powders is on the nano-scale. In detail, it was estimated at 16.3, 7.7, and 11.1 nm for N-doped, Fe-doped, and Fe,N-doped TiO_2_ particles, respectively. Therefore, nanostructured powders were produced with a lower average crystallite size than those recorded in the literature [34,35]. In this study, aiming to propose a stimuli responsive drug delivery system, nano-sized particles are preferable. A reduced size on the nano-meter scale is related to enhanced activity of the TiO_2_ powders and efficient embedding in the IP network microgel pNiPam-co-PAA.

The morphology of the produced doped TiO_2_ powders was examined by FESEM. The images of the doped TiO_2_ powders are shown in Figure 2.

At high magnification images (Figure 2d–f), it is possible to detect grains in the range of nano-meters, for N-TiO_2_ is ~14 nm, for Fe-TiO_2_ is ~10 nm, and, lastly, for Fe,N-TiO_2_ is ~11 nm. These are in accordance with our XRD previously mentioned results. In the case of the Fe dopant, a more heterogeneous structure is observed (Figure 2b,c), while a small percentage of agglomeration is observed mainly in the case of co-doped powders (Figure 2c).

Figure 3 illustrates the Raman spectra of the as-prepared powders.

All the peaks observed correspond to the Raman fundamental modes of pure anatase crystal phase, located at 143 (E_g_(1)), 197 (E_g_(2)), 395 (B_1g_(1)), 514 (A_1g_), and 640 (E_g_(3)) cm^−1^. No other crystalline form of TiO_2_ was detected, as well as no peak corresponding to iron oxide. A small red-shift is observed at the main peak of 143 cm^−1^ (see inset of Figure 3), under the presence of the Fe dopant. This is possibly associated with the doping effect, which leads to an alteration of the TiO_2_ lattice. This is a change in grain size or a presence of defects [57]. These findings are in agreement with the XRD results of this study, as well as with the relevant literature for Raman analysis [57,58].

Figure 4 presents the FTIR spectra of all the as-produced chemically modified TiO_2_ nano-powders.

In all the samples, a broad band at 3050–3450 cm^−1^ and a band at 1650 cm^−1^ are assigned to the stretching and bending vibration of O–H, respectively. These hydroxyl groups originate from water absorbed on the surface of the TiO_2_ samples. Their presence may increase the photocatalytic activity of the synthesized powders, which results in the formation of the Reactive Oxygen Species that could oxidize organic pollutants or cancer cells [57,59]. At around 2300 cm^−1^, the peaks are attributed to the C-residuals, due to the use of organic solvents at the synthesis process. The vibrational peak at 670 cm^−1^ corresponds to the Ti–O stretching mode of Ti–O–Ti [60].

In order to examine the surface and the chemical state of the elements present in the as-prepared doped TiO_2_ nano-powders, XPS analysis was performed. Figure 5a, Figure 6a and Figure 7a depict the wide survey spectrum of N-TiO_2_, Fe-TiO_2_, and Fe,N-TiO_2_ nanoparticles, respectively. All the peaks expected due to the chemical synthesis followed are detected, i.e., the Ti2p (TiO_2_ state), O1s (O in TiO_2_ state), C1s (hydrocarbon state), in accordance with published relative literature [35]. Figure 5b shows the peak of N1s element in high resolution, which corresponds to the chemical state of N-Ti-O bonds and is located at 401.07 eV [30,59]. Figure 6b presents the peak of Fe2p3/2, found at 710.81 eV [35], attributed to Fe^+3^, which results in the formation of Fe_2_O_3_ [34,59]. Moreover, Figure 7b displays the Fe peak at 709.6 eV and Figure 7c displays the N1s peak at 399.4 eV. The Fe2p3/2 corresponds to the FeO state and the N1s to Ti-N bonds [61,62]. N1s peak of N-doped powder has an atomic concentration of 0.34 ± 0.07%, Fe2p3/2 of Fe-doped 0.21 ± 0.03%, and, lastly, Fe2p3/2 and N1s of Fe,N-doped powder 0.25 ± 0.02% and 0.83 ± 0.25%, respectively. Ti2p peak of all doped TiO_2_ exhibited a surface atomic concentration of 20% to 25%.

By using Uv-Vis spectroscopy, the *E_g_* of the nano-powders was estimated. First, the reflectance of the powders at Kubelka-Munk (K-M) units is measured. The application of the K-M method is based on the equation below.(2)$$F(R)=\frac{{\left(1-R\right)}^{2}}{2R}$$ where R is the reflectance. This method is usually applied when the samples measured exhibit high absorbance or light scattering [63]. Figure 8 depicts the change in the reflectance of the powders within the light spectrum.

By using the Tauc’s equation, a semiconductor’s *E_g_* is calculated.(3)$$ahv=A{\left(hv-E_{g}\right)}^{n}$$ where *E**_g_* is the energy band gap, *α* is the absorption coefficient, *A* is a constant, and *n =* 1/2 for direct [57]. By applying the K-M method, the band gap energy can be obtained by extrapolating the linear region of the spectra (F(R)hv)12 vs. *hv*, as presented in Figure 9.

The measured band gaps of the commercial non doped Evonik P25 and the produced TiO_2_ nano-powders (inset of Figure 9) revealed that there is a decrease of *E_g_* compared to pure anatase, due to doping with N, Fe, and Fe, N. This may be related to a better photocatalytic activity under visible light irradiation, since the catalyst needs lower energy in order to be activated. The N-doped [30] and Fe,N-doped [34] powders’ results are in close agreement with similar studies. The observed redshift of *E_g_* is attributed to a combination of both substitutional and interstitial doping of N [30] and to the overlapping conduction bands of TiO_2_ and 3d electrons of Fe^+3^ ions [35,57]. The reflectance of Fe-doped powder slightly differs from what is reported by literature [35], which is possibly associated with the presence of iron oxide, as observed by the XPS data. Therefore, the band gap peak of iron oxide overlaps the TiO_2_ band gap peak, which results in a lower value but not, necessarily, in better photocatalytic performance [64].

### 3.2. Characterization of the IP Network Microgel pNipam-co-PAA and the Composite Materials

The hydrodynamic diameter (Dh) of the particles of the IP network microgel pNipam-co-PAA was determined via DLS over a temperature range of 25 °C ≤ T ≤ 43 °C with a step of 0.5 °C and at a constant pH value equal to 6. All measurements were conducted by using a red laser line operating at 633 nm. Additionally, 1 mL of microgel solution was placed in a DTS1070 capillary cell (Malvern Instr., Malvern, UK). The temperature equalization time was set at 900 s before each measurement, due to the slow temperature response of the microgel pNipam-co-PAA. VPTT is defined as a sharp but continuous transition in the volume of the hydrodynamic diameter at the swollen and collapsed phase [40,65].

The DLS data depicted in Figure 10 reveals that the collapsed hydrodynamic diameter of the microgel is ~400 nm and the swollen diameter is ~720 nm. The VPTT is calculated at 37 °C, which indicates a shift to higher values, compared to that reported for pNipam [39]. This is preferable for operating in a human body environment. This shift is accomplished with the proper co-polymer selection and the optimization of the time interval between the additions of the two monomers of the co-polymerization during the microgel synthesis process [40]. In a water medium, below the VPPT, the chains are soluble due to hydrogen bonds forming between water and amine chains and, over the VPTT, the water is discarded from the microgel and the volume shrinks [36,39,66].

The morphology of the produced composite IP network microgel/doped-TiO_2_ particles was examined by TEM microscopy and is presented in Figure 11. Figure 11a shows embedded P25 Evonik particles at the surface and in the core of the polymeric particles, which is in agreement with previous relative studies [41,43]. In order to ensure the embedding of the doped TiO_2_ nanoparticles in the core of the polymeric network, TEM images tilting from a 0° to −45° angle are demonstrated for N-doped composites (Figure 11b,e). Inorganic particles are observed in the exact same position, which the arrows pinpoint. Figure 11c shows the Fe-TiO_2_ composite particles, the cross-linked network structure of the microgel, and the embedded particles. In the dark field image of the same sample at higher magnification (Figure 11f), the inorganic particles are more discernable and exhibit a size of approximately 10–20 nm, while those of composite particles is in the range of ~300–400 nm. It is worth mentioning that, these findings verify the size of the doped-TiO_2_ particles as recorded by XRD and FESEM data of this study. The same embedding type of the inorganic particles in the IP network was, also, observed for the Fe,N-TiO_2_ powders (Figure 11d). The asymmetry between the shape (drop or lobe) of the polymeric particles is a result of the solvent drying between particles during the TEM sample preparation [41].

In Figure 12, the Raman spectrum of pNipam-co-PAA/N-TiO_2_ composite particles is illustrated. All the peaks observed in the region of 130–650 cm^−1^ are attributed to the Raman fundamental modes of pure anatase crystal phase in agreement with those reported for doped TiO_2_ nanoparticles (see Figure 3). This finding verifies the presence of N-doped TiO_2_ particles within the IPN microgel that has been also observed in TEM results (see Figure 11).

In the frequency range of 2850–3050 cm^−1^ (see inset of Figure 12), four peaks are assigned to the different C-H stretching modes of the Nipam molecule in the hydrate state, according to previous related studies: at 2880 cm^−1^, the symmetric stretching of CH_3_, at 2920 cm^−1^, 2945 cm^−1^ symmetric and antisymmetric stretching of CH_2_, respectively, and at 2988 cm^−1^, the antisymmetric stretching of CH_3_ [67,68].

The XPS survey spectrum of pNipam-co-PAA/N-TiO_2_ composite particles depicted in Figure 13a demonstrates peaks at 285.0, 399.7, 458.5, and 529.8 eV associated with C1s, N1s, Ti2p, and O1s, respectively. The N1s peak is attributed to the nitrogen in pNipam (see Figure 13b), as well as the corresponding peak of C1s in accordance with related studies [44,69]. Additionally, the high resolution XPS spectrum of Ti2p exhibited the expected peak separation between Ti2p1/2 (located at 464.4 eV) and Ti2p3/2 (located at 465.5 eV), in agreement with reported literature [70]. The existence of TiO_2_ in the as-produced composite particles is also verified by the TEM and Raman data of this study. As far as the surface concentration of the N-doped TiO_2_ particles embedded in the polymer microgel is concerned, the Ti at percentage analysis is 12.4%, which is a value expected by taking into consideration the loading of the particles in the IP microgel (i.e., 50% mass).

### 3.3. Photocatalytic Results

The photocatalytic behavior of the as-produced doped-TiO_2_ and composite particles was assessed by testing the photocatalytic degradation of a solution of the MB pollutant. All the photocatalytic tests were conducted at room temperature and pH=6. Figure 14a illustrates the photocatalytic behavior of all the doped-TiO_2_ powders, as well as of the non-doped Evonik P25. Figure 14b displays the photocatalytic kinetics of the same samples, respectively. In Figure 14a, the photolysis of the MB pollutant is included, together with its degradation rate without the presence of light irradiation.

It is clear that the N-doped powder almost fully degrades the pollutant. At 150 min of visible light irradiation, MB is 95% degraded for N-doped, 47% for Fe, N-doped, and 40% for Fe-doped. For all these percentages, a 25% is attributed to the photolysis of the pollutant [13], while, in the absence of the photocatalyst, the photodegradation of MB in the dark showed no difference. Therefore, the 30% degradation of the Evonik P25 is attributed mainly to the photolysis of MB. Additionally, there was a maximum of 10% adsorption of MB in the tested powders when kept in the dark for 1 h. This percentage was subtracted from the data of Figure 14a. The chemical modification of TiO_2_ results in a higher photocatalytic degradation of the pollutant, compared to the non-doped commercial powder. This is due to the dopant suppressing the recombination of the photo-generated electrons and holes [57,71]. In addition, when TiO_2_ is doped, secondary energy states will be formed between the valence and conduction band, which alters the band gap value as verified by the Uv-Vis data of this study for N and Fe dopants. Nonetheless, the low photocatalytic activity of the Fe-doped TiO_2_ could be associated with the excessive concentration of iron doping. The photocatalytic activity of Fe-doped TiO_2_ is strongly dependent on the dopant concentration since Fe^3+^ ions can serve as electron hole recombination centers, as well as the mediator of interfacial charge transfer [55]. Moreover, the accumulation of the dopant covers part of the photocatalyst’s surface and, thus, reduces the number of surface-active sites [57]. Additionally, it has been reported that the visible light photocatalytic activity of Fe-doped TiO_2_ is limited by the formation of iron oxide (see corresponding XPS results) on the surface of the nanoparticles [72]. Overall, the observed photocatalytic performance is consistent with relative studies of Fe [57] and Fe,N [71] doped-TiO_2_ particles.

The Langmuir-Hinshelwood model of pseudo-first kinetics order is generally fitted in the photocatalytic degradation, by following the equation below [73].(4)−ln(CC0)=kappt where C_0_ is the initial concentration of MB equal to 0.6∙× 10^−5^ M, C is the concentration of MB at irradiation time t and k_app_ is the pseudo-first order rate constant. Table 1 presents the calculated photoinduced degradation rate constant and the R^2^ coefficient of the linear regression fitting for all the TiO_2_ tested samples. All R^2^ values indicate that the linear kinetic is a rather suitable photocatalytic model for all the powders. The estimated $k_{\mathrm{app}}$ values demonstrate that the photocatalytic decomposition of MB is higher when N-doped TiO_2_ is used.

Figure 15a illustrates the photocatalytic degradation of MB for all the embedded doped-TiO_2_ powders in the IP network microgel pNipam-co-PAA, as well as that of Evonik P25. Figure 15b presents the kinetic results (adopting the Langmuir-Hinshelwood model). The pure IP microgel photocatalytic behavior was also studied and it revealed no influence on the degradation of MB (Figure 15). A similar photocatalytic behavior to the one reported for the TiO_2_ powders (see Figure 14a) is observed for the composite particles. At 150 min of visible irradiation, MB is degraded 75.5% for N-doped, 33% for Fe,N-doped, and 27% for Fe-doped. Therefore, the 23% degradation of the Evonik P25 is attributed mainly to the photolysis of MB. Therefore, it appears that the Fe-doped composite does not exhibit a photocatalytic effect on the pollutant, since its degradation percentage is close to that of the MB photolysis. However, N-doped composites are the ones demonstrating an enhanced photocatalytic behavior. The degradation rate decrease of these composites compared to the pure powder (Figure 14a) could be attributed to a very low concentration of TiO_2_ nanoparticles in the microgel, which is an outcome of the 50% (mass) TiO_2_ mixing in the microgel for forming the composite particles, which was also supported by the XPS surface at a percentage concentration analysis. The adsorption of the MB pollutant in the composite particles was about 50% to 52%, which is due to the strong interaction between MB dye and pNipam microgels [74]. This result was expected, as microgels based on pNipam are used for their ability to remove different contaminants from water [75]. This adsorption percentage is subtracted from the data of Figure 15a.

Table 2 presents the calculated photoinduced degradation rate constant (k_app_) and R^2^ coefficient of the linear regression fitting for all the composite particles. The linear kinetic model is satisfactorily fitted in all experimental data, as depicted by the R^2^ values. As above mentioned, N-doped TiO_2_ particles embedded in the IP microgel exhibit the best photocatalytic performance.

A schematic diagram illustrating the basic principle of doped TiO_2_ photocatalyst with the presence of an organic pollutant in accordance with published data [30,34,57] is presented in Figure 16. When photons from a light source hit the surface of a TiO_2_ nanoparticle, electrons are excited from the valance to conduction band, which leaves behind positively charged holes. The negatively charged electrons interact with molecular oxygen, whereas the positive holes react with humidity water molecules, which form superoxide anion radicals and hydroxyl radicals or ROS (reactive oxygen species). These ROS play a vital role in the photocatalysis mechanism, since they oxidize organic pollutants or cancer cells gradually to form simpler inorganic compounds. The incorporation of N into the TiO_2_ lattice leads to the formation of a new mid-gap energy, which decreases the band gap of TiO_2_ and shifts the optical absorption to the visible light region [30]. For the Fe doping, it has been demonstrated [57] that the doping of Fe^3+^ ions in a TiO_2_ lattice decreases its band gap ascribed to the overlap of the conduction band due to the Ti (*d* orbital) of TiO_2_ and the metal (*d*-orbital) orbital of Fe^3+^ ions. Furthermore, it is generally accepted that the doping of Fe^3+^ ions induces the formation of new electronic states (Fe^4+^ and Fe^2+^) that span across the band gap of TiO_2_. Moreover, these electronic states may act as electron and hole trapping sites, which result in the reduction of the recombination of the electron and hole, and, ultimately, enhance photocatalytic activity. However, it was proven that the photoactivity of the Fe/TiO_2_ catalyst is dependent on the way of preparation and the amount and state of iron [76]. In case of Fe,N-doped, the co-doping with nitrogen and Fe^3+^ induces a red shift of the band gap, compared to bare TiO_2_. Therefore, the nitrogen and Fe^3+^ ion doping induced the formation of new states closed to the valence band and conduction band, respectively. The co-operation of the nitrogen and Fe^3+^ ion leads to the narrowing of the band gap, as well as inhibits the recombination of the photogenerated electron and hole [71]. The suitable concentration range of Fe ion can trap the photogenerated electron, while the nitrogen can trap part of photogenerated holes, which enhances the utilization efficiency of photogenerated electrons and holes [77].

### 3.4. Biological Effect

#### 3.4.1. Cell Proliferation

Growth rates allow us to evaluate the effects of doped TiO_2_ particles on the cell proliferation of both MCF-7 and MDA-MB-231 cells. Cells were incubated with increasing concentrations of doped TiO_2_ dispersions and the cell number was recorded as a function of time for the three different photocatalysts upon irradiation with visible light. The range of concentration of dispersions tested was 0–1 mg/mL. The optimized concentration of 0.8 mg/mL is presented, which allowed the detection of a significant difference on the biological effect among the two different cell lines. As a positive control of the experiments, cells treated for 24 h with cisplatin (1 mg/mL) were used. As an extra internal negative control, the cells treated with visible light without TiO_2_ were considered, in order to ensure that the effect is not relevant to the irradiation itself. Thus, the phenomenon of hyperthermia as a possible mechanism, which results in the inhibition of cell proliferation, can be excluded.

As shown in Figure 17a, for the MCF-7 (non-metastatic cancer) cells, no significant effect on cell proliferation was observed, even after the photo-activation with visible light. On the contrary, cell proliferation of the highly malignant MDA-MB-231 gradually decreased in the presence of photo-activated N-doped TiO_2_ (Figure 17b). Additionally, Fe-doped and Fe,N-doped TiO_2_ nanoparticles are proven to be less effective than N-doped TiO_2_, while the same dose of 0.8 mg/mL did not affect the MCF-7 cell population.

These preliminary results indicate that the highly malignant MDA-MB-231 cancer cells are more susceptible to the inhibition of cell proliferation or, possibly, to cell death when exposed to photo-activated N-doped TiO_2_ nanoparticles compared to MCF-7 cells, which are characterized by low metastatic potential. Thus, there is a cell-dependent cell toxicity [10,46]. This behavior could be explained by considering the different nature of these cell lines and possibly the different membrane receptors, which could, consequently, interact with the TiO_2_ nanoparticles differently. This leads to a different biological effect [10,78]. The protein composition of the cell membranes and the manner of membrane proteins and TiO_2_ nanoparticles’ interactions may lead to insignificant influence of TiO_2_ on MCF-7 cells. Additionally, MDA-MB-231 cells demonstrate stem cell characteristics, such as high expression of cancer stem cells (SCCs) markers CD44/CD133 and high activity of aldehyde dehydrogenase (ALDH), which is involved in stem cell self-protection, compared to MCF-7 [79]. Furthermore, the low toxicity of doped TiO_2_ on MCF-7 cells could be attributed to the fact that MCF-7 cells are very resistant, because they contain a xenobiotic transporter (BCRP), which plays a crucial role in the multi-drug resistance.

Moreover, N-doped TiO_2_ particles had a higher biological effect compared to the effect of Fe-doped or Fe, N-doped TiO_2_ nanoparticles. This could be associated with the low solubility of iron in biological systems [80,81]. In addition, there are some studies indicating that iron could protect against oxidative stress-induced cellular damage [82]. Thus, the presence of iron may inhibit the potential of TiO_2_ nanoparticles to induce cell death, by protecting the cell population.

Figure 18a reveals there is no significant effect on cell proliferation in the case of the non-metastatic MCF-7 cells in the presence of the embedded N-doped, Fe-doped, or Fe,N-doped TiO_2_ nanoparticles in the IP network microgel pNipam-co-PAA, even upon irradiation with visible light. This was quite predictable, since the pure doped-TiO_2_ nanoparticles also do not affect the cell proliferation of these cells (Figure 17a). In addition, it is important to mention that pNipam-co-PAA is a biocompatible material and that, in the presence of increasing concentrations of pNipam-co-PAA, both cell lines are totally unaffected.

However, there is a clear effect of the embedded N-doped TiO_2_ on MDA-MB-231 cells, since there is a ~15% decrease in the cell number (Figure 18b). This less noticeable effect compared to doped TiO_2_ nanoparticles is reasonable, since the loading of TiO_2_ in the IP network microgel pNipam-co-PAA is estimated to be around 50% mass fraction. This implies that, at the concentration of 0.8 mg/mL of the composite particles, only half of the amount of doped TiO_2_ particles are present to act as a catalyst. Thus, it is rational to forecast an inhibition of cell proliferation in the presence of embedded N-doped TiO_2_ particles, even though pure nanoparticles exhibit the best photo-induced biological effect (Figure 17b). This is a very promising result because it seems that the embedding in the IP network microgel pNipam-co-PAA allows the control of the release of nanoparticles in the biological system, which, thus, decreases the concentration of N-doped TiO_2_ nanoparticles needed to cause a significant effect on cell proliferation.

Therefore, it is rational to present the biological effect of the concentration of 0.4 mg/mL of the doped particles (see Figure 19), in order to allow a straight comparison between the composites (Figure 18) and doped nanoparticles. In Figure 19a,b, there is no remarkable effect on both cell types in the presence of N-doped, Fe-doped, and Fe, N-doped TiO_2_ nanoparticles on both MCF-7 and MDA-MB-231 cells. As it is mentioned, there is a clear effect of the embedded N-doped TiO_2_ on MDA-MB-231 cells (Figure 18b) at the same concentration. Therefore, it seems that this polymeric microgel efficiently releases the N-doped nanoparticles upon or very close to the cell membranes, which inhibits the cell proliferation of MDA-MB-231 cells, while the non-embedded nanoparticles can disperse everywhere in the cell culture, by decreasing their efficacy. This decrease in the dose of the TiO_2_ catalyst that simultaneously affects the metastatic cancer cells could be considered an optimized factor of the proposed photodynamic therapy of this study.

#### 3.4.2. Oxidative Stress Detection

Glutathione (GSH) is the smallest intracellular protein thiol molecule in the cells, preventing cell damage caused by ROS. Glutathione exists in reduced (GSH) and oxidized (GSSG) states [83]. Reduced glutathione (GSH) is a major antioxidant that provides reducing equivalents for the glutathione peroxidase (GPx) catalyzed reduction of lipid hydroperoxides to their corresponding alcohols and hydrogen peroxide to water [84]. In the GPx catalyzed reaction, the formation of a disulfide bond between two GSH molecules generates oxidized glutathione (GSSG). In healthy cells, under normal conditions, the most prevalent form of the glutathione pool is the reduced form (GSH) [85]. However, during normal aging, in neurodegenerative diseases, and, when the cells are exposed to increased levels of reactive oxygen species, GSSG accumulates and the ratio of GSSG to GSH increases. An increased ratio of GSSG-to-GSH is an indication of oxidative stress and it is considered to be one of the most reliable techniques for the detection of a redox potential [86].

As shown in Figure 20a, there are no detectable changes in the GSSG-to-GSH ratio, in the presence of the embedded or non-embedded N-doped, Fe-doped, and Fe,N-doped TiO_2_ nanoparticles on MCF-7 cells. However, there is a clear increase in the GSSG-to-GSH ratio in the presence of the photo-activated embedded N-doped TiO_2_ on MDA-MB-231 cells, while there is a lower decrease in the presence of photo-activated N-doped TiO_2_, compared to untreated cells (Figure 20b). Cells treated with 1 mM H_2_O_2_ were considered as a positive control of these series of experiments. In addition, it is noteworthy that there was no indication of oxidative stress in the presence of microgel in both cell lines. These results indicate that the possible mechanism of the cell inhibition previously presented is mediated by oxidative species generation. Further systematic investigation of these findings is mandatory in order to gain a better insight on the corresponding molecular mechanism associated with the anticancer behavior of the TiO_2_ particles and composites.

## 4. Conclusions

Nano-powders of N-doped, Fe-doped, and Fe, N-doped TiO_2_ were synthesized by the sol-gel method. The photocatalytic activation of the produced TiO_2_ powders was broadened under the visible light spectrum due to doping with Fe and N elements. The doping effect was verified by the XPS analysis of the surface of the powders. The dominant crystal phase of the doped TiO_2_ powders was anatase, with decreased *E_g_* values compared to non-doped Evonik P25. The crystalline size of the produced catalysts was, on the nano-scale, between 10 to 20 nm. Photocatalytic tests using MB pollutant revealed the enhanced photocatalytic activity in visible light of the N-doped TiO_2_ particles.

A stimuli responsive IP network microgel of pNipam-co-PAA was synthesized by a precipitation polymerization process, which presents a VPTT at 37 °C. The as-prepared TiO_2_ nanoparticles were embedded in the IP network microgel, which results in the formation of pNipam-co-PAA/(co)doped-TiO_2_ composite particles, aiming to produce a thermo-responsive drug delivery system operating in environmental conditions in the human body. The cytotoxicity tests of TiO_2_ nanoparticles using visible light irradiation have shown a decrease in cell proliferation on MDA-MB-231 cells, especially for the N-doped TiO_2_ nanoparticles. For the MCF-7 (non-metastatic) cancer cells, no significant effect on cell proliferation was observed, which pinpoints a selectivity of the synthesized types of particles toward specific cancer cells. The investigation of the mechanism provoking this selectivity is an interesting field for further future research. A less noticeable effect of the composite embedded N-doped TiO_2_ particles on the MDA-MB-231 cell proliferation was revealed when compared to doped TiO_2_ nanoparticles. Our experimental findings indicate that the possible mechanism, which leads to the inhibition of cell proliferation is the ROS generation. Moreover, it should be highlighted that the anti-cancer behavior is in line with those of photo-induced MB degradation tests for the TiO_2_ and composite particles. These early-stage-research results are promising for promoting the development of a photodynamic cancer therapy under visible light irradiation, since the embedding of doped-TiO_2_ in an IP network microgel pNipam-co-PAA allows the control of the catalyst’s release in the biological system while simultaneously maintaining its photo-induced activity.

## Figures and Tables

**Figure 1 materials-12-02589-f001:**
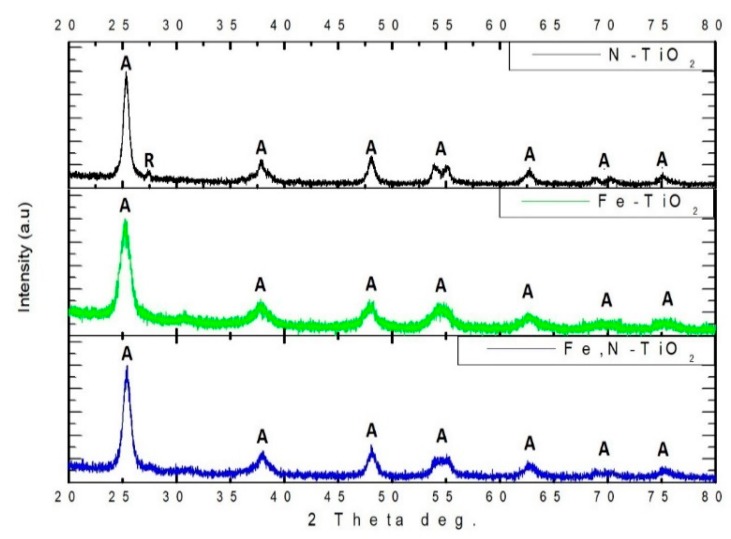
XRD patterns of N-doped TiO_2_, Fe-doped TiO_2_, and Fe,N-doped TiO_2_ powders.

**Figure 2 materials-12-02589-f002:**
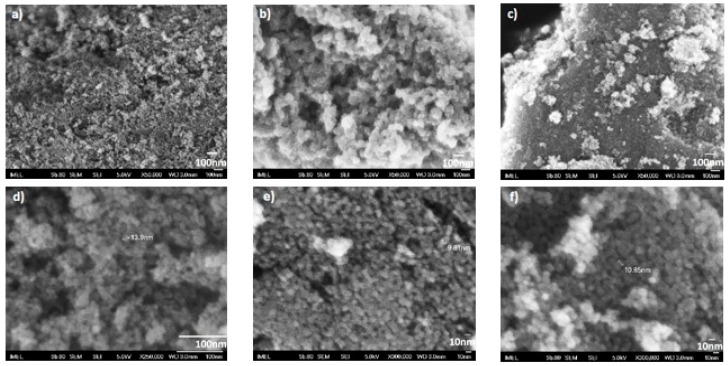
FESEM images of (**a**) N-doped TiO_2_, (**b**) Fe-doped TiO_2_, (**c**) Fe,N-doped TiO_2_ at ×50,000 magnification, (**d**) N-doped TiO_2_ at ×250,000 magnification, (**e**) Fe-doped TiO_2_, and (**f**) Fe,N-doped TiO_2_ at ×300,000 magnification.

**Figure 3 materials-12-02589-f003:**
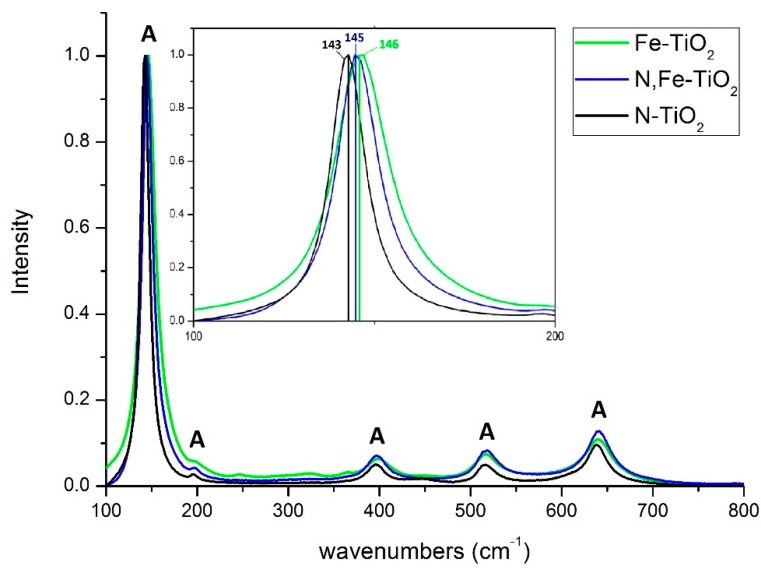
Raman spectra of doped-TiO_2_ nano-powders. The inset depicts the region of the main Raman peak in detail.

**Figure 4 materials-12-02589-f004:**
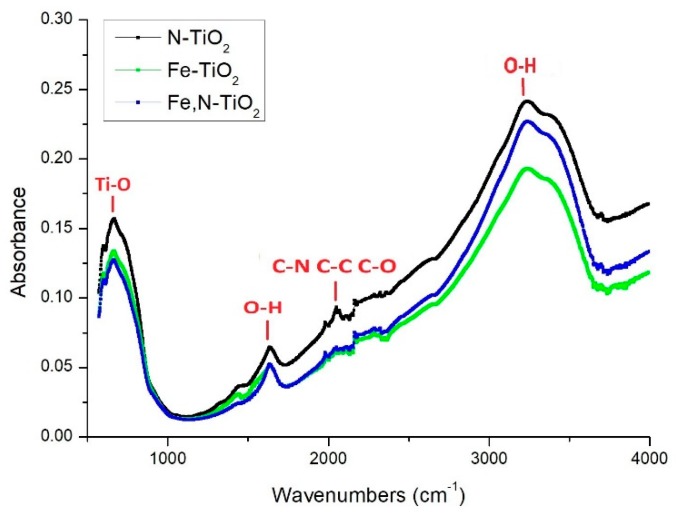
FTIR spectra of doped-TiO_2_ nano-powders at 25 °C.

**Figure 5 materials-12-02589-f005:**
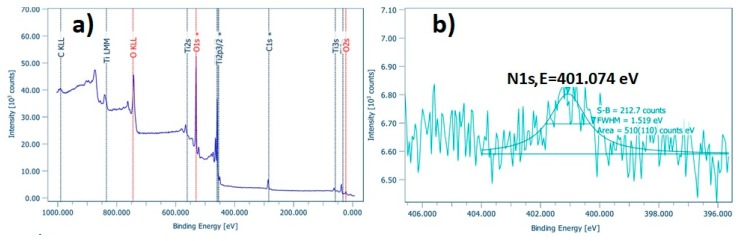
(**a**) Wide survey and (**b**) high resolution of N1s XPS spectra of N-TiO_2_ powder.

**Figure 6 materials-12-02589-f006:**
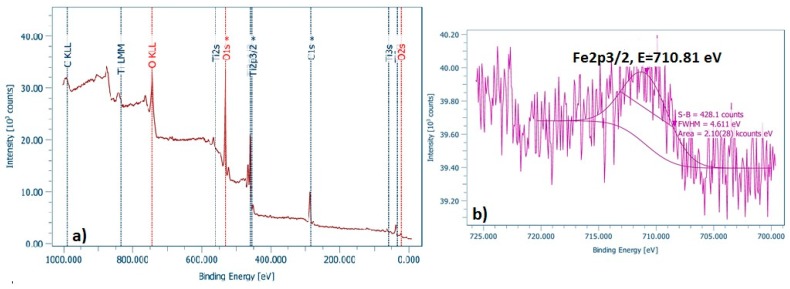
(**a**) Wide survey and (**b**) high resolution of Fe2p3/2 spectra of Fe-TiO_2_ powder.

**Figure 7 materials-12-02589-f007:**
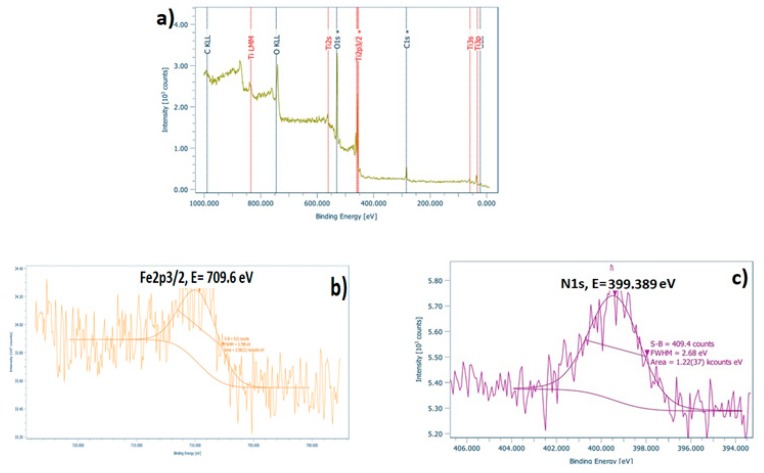
(**a**) Wide survey, (**b**) high resolution of Fe2p3/2, and (**c**) high resolution of N1s spectra of Fe,N-TiO_2_ powder.

**Figure 8 materials-12-02589-f008:**
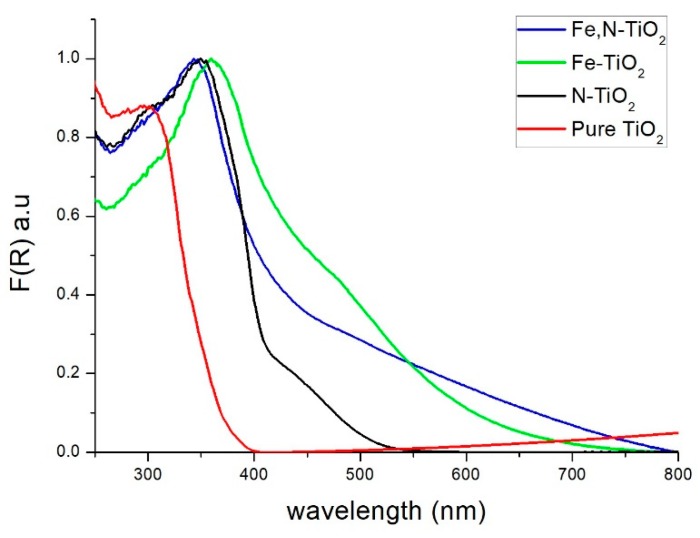
Variation of F(R) reflectance vs. wavelength for Evonik P25 and the three doped powders.

**Figure 9 materials-12-02589-f009:**
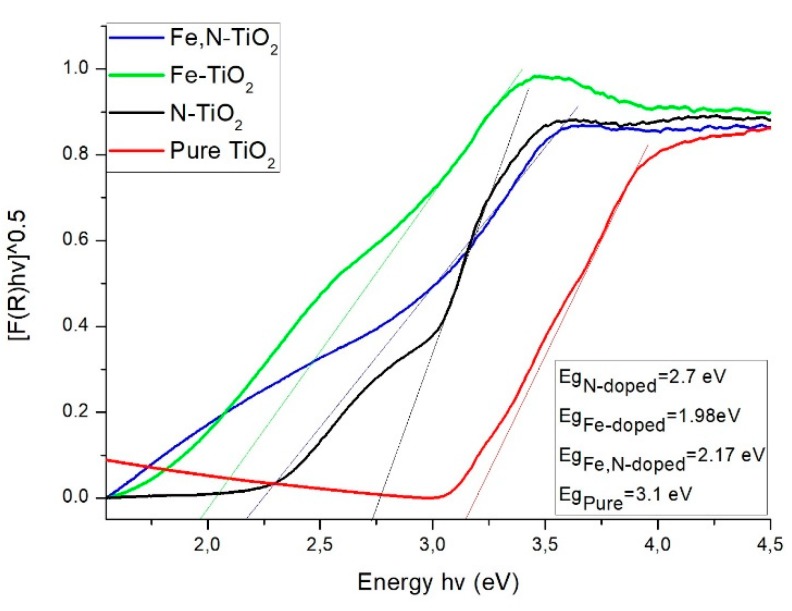
Optical band gap energy of Evonik P25 and doped-TiO_2_ nano-powders.

**Figure 10 materials-12-02589-f010:**
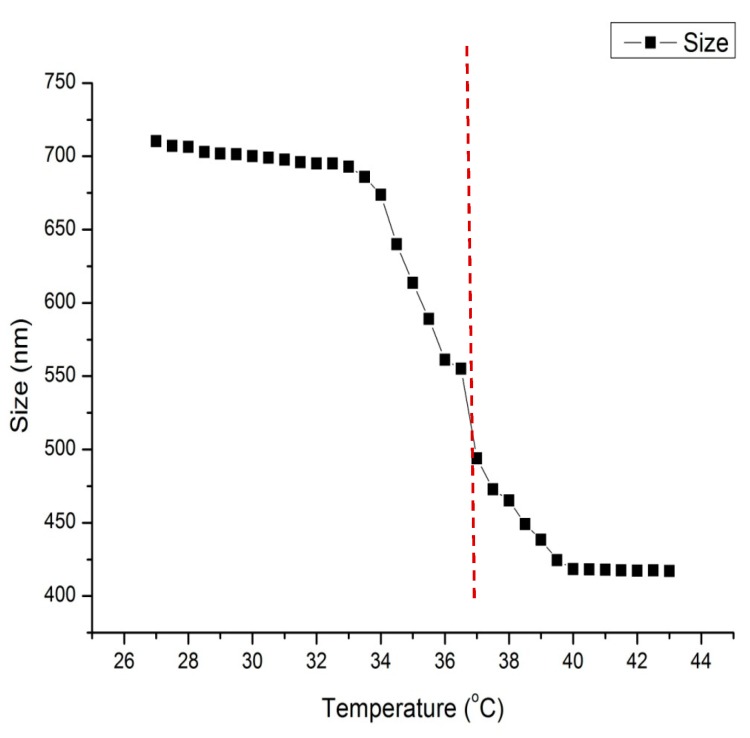
Variation of hydrodynamic diameter (size) of microgel pNipam-co-PAA to temperature using DLS. The pH of the suspensions is equal to 6. The dash line underlines the VPTT.

**Figure 11 materials-12-02589-f011:**
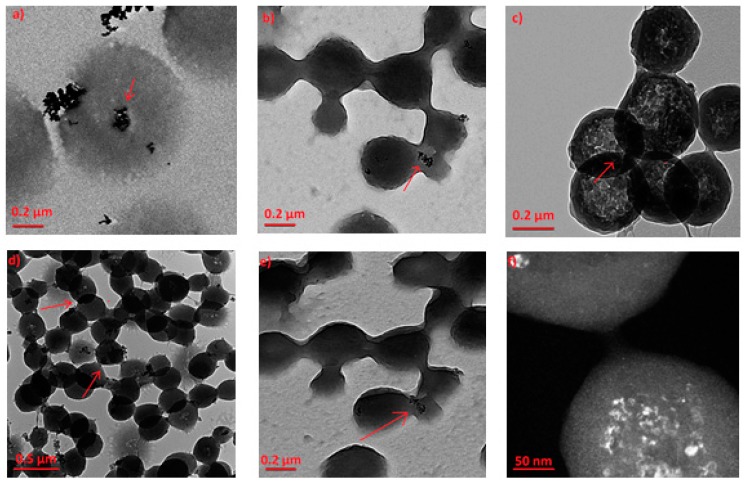
TEM images of the embedded TiO_2_ powders in the IP network microgel pNipam-co-PAA for (**a**) Evonik P25, (**b**) N-doped at 0°, (**c**) Fe-doped, (**d**) Fe,N-doped, (**e**) N-doped −45°, and (**f**) Fe-doped (dark field).

**Figure 12 materials-12-02589-f012:**
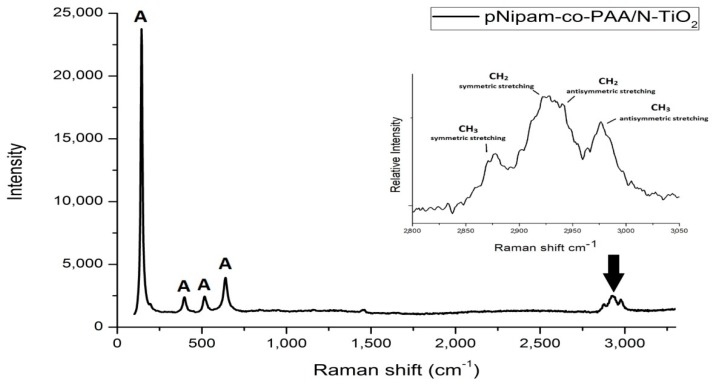
Raman spectrum of pNipam-co-PAA/N-TiO_2_ composite particles. The inset depicts the Raman spectrum of composite particles zoomed in on the high frequency region.

**Figure 13 materials-12-02589-f013:**
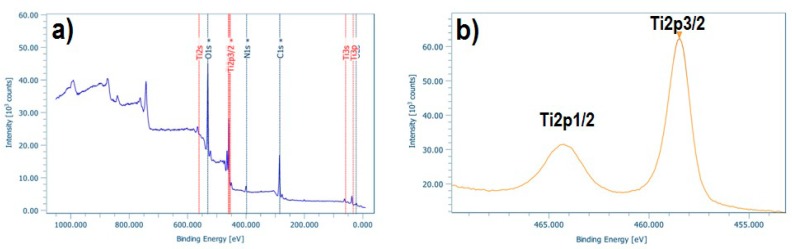
XPS spectra of pNipam-co-PAA/N-TiO_2_ composite particles (**a**) wide survey and (**b**) high resolution of Ti2p.

**Figure 14 materials-12-02589-f014:**
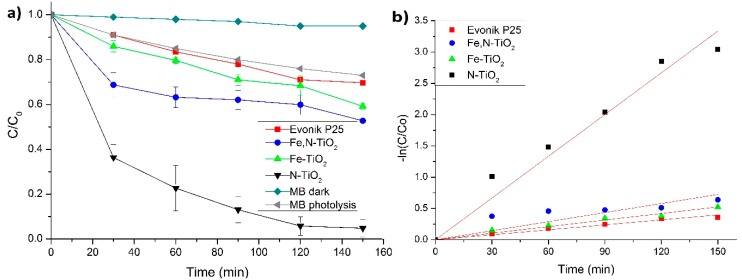
(**a**) Degradation curves of MB for doped and non-doped powders vs. time, under visible light irradiation. The photolysis of MB and its degradation in the dark are included. (**b**) Photocatalytic kinetics of the same samples, following a linear pseudo-first order model.

**Figure 15 materials-12-02589-f015:**
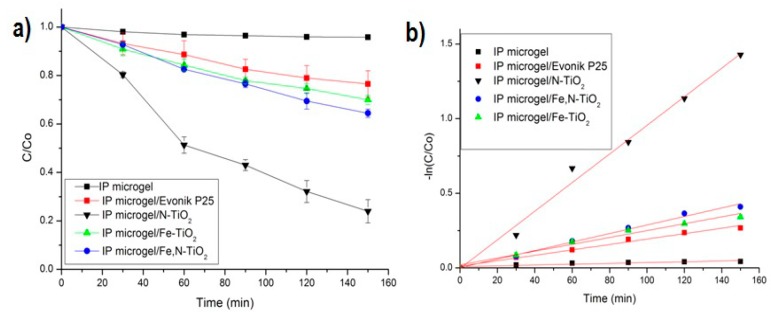
(**a**) Degradation curves of MB for all the composite particles vs. time, under visible light irradiation. The photolysis of MB and its degradation in dark are included. (**b**) Photocatalytic kinetics of the same samples, following a linear pseudo-first order model. The pure IP network microgel is also included in both schemes.

**Figure 16 materials-12-02589-f016:**
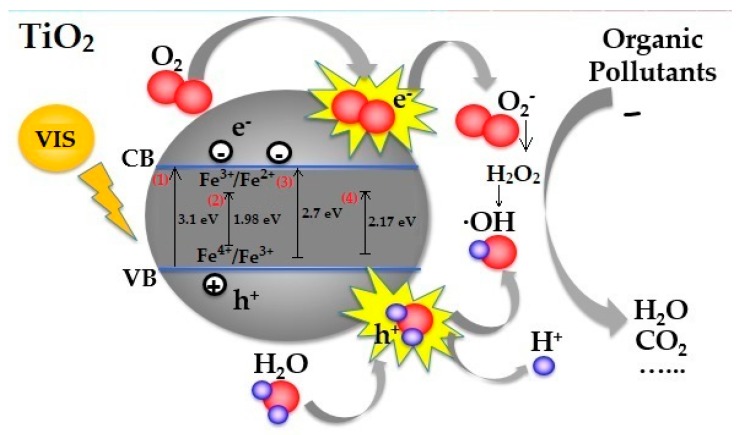
Schematic representation of the possible proposed photocatalytic mechanism of the: (1) non-doped TiO_2_, (2) Fe-doped TiO_2_, (3) N-doped TiO_2_, and (4) Fe,N-doped TiO_2_ particles under visible light irradiation.

**Figure 17 materials-12-02589-f017:**
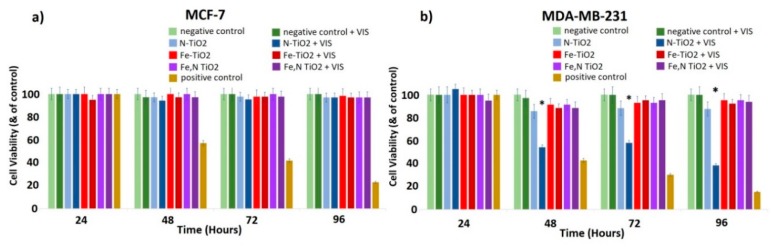
Effect of N-doped, Fe-doped, and Fe, N-doped TiO_2_ nanoparticles (0.8 mg/mL) on (**a**) MCF-7 and (**b**) MDA-MB-231 breast cancer epithelial cell proliferation. Cells treated for 24 h with cis-platin (1 mg/mL) were used as a positive control of the experiments, untreated cells as a negative control, and cells treated with visible light without TiO_2_ as an additional negative control. * *p* < 0.05 compared to the negative control (untreated) cells. Data represent means ± standard deviation from five independent experiments.

**Figure 18 materials-12-02589-f018:**
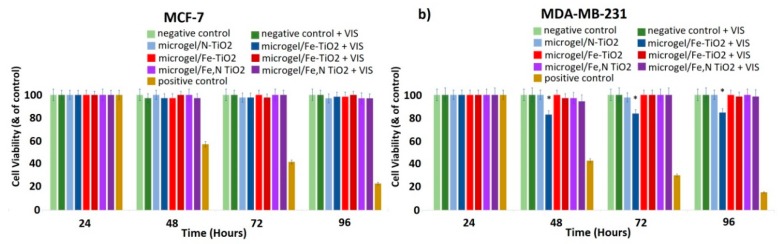
Effect of embedded N-doped, Fe-doped, and Fe,N-doped TiO_2_ nanoparticles (0.8 mg/mL)on (**a**) MCF-7 and (**b**) MDA-MB-231 breast cancer epithelial cell proliferation. Cells treated for 24 h with cis-platin (1 mg/mL) were used as a positive control of the experiments, untreated cells used as a negative control, and cells treated with visible light without TiO_2_ as an additional negative control. * *p* < 0.05 compared to negative control (untreated) cells. Data represent means ± standard deviation from five independent experiments.

**Figure 19 materials-12-02589-f019:**
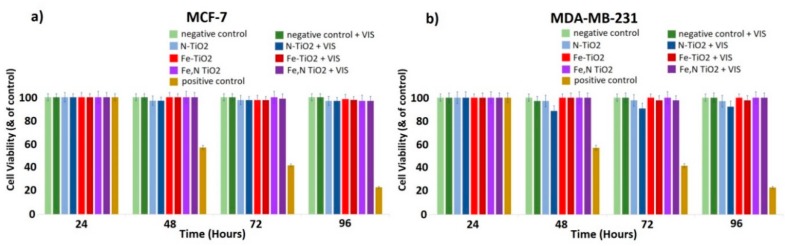
Effect of N-doped, Fe-doped, and Fe, N-doped TiO_2_ nanoparticles (0.4 mg/mL) on (**a**) MCF-7 and (**b**) MDA-MB-231 breast cancer epithelial cell proliferation. Cells treated for 24 h with cis-platin (1 mg/mL) were used as a positive control of the experiments, untreated cells were used as a negative control, and cells were treated with visible light without TiO_2_ as an additional negative control. Data represent means ± standard deviation from five independent experiments.

**Figure 20 materials-12-02589-f020:**
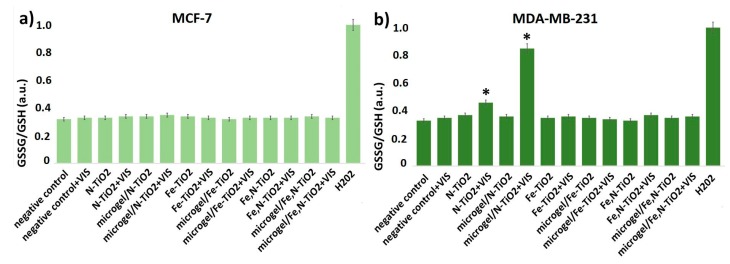
Effect of embedded and non-embedded N-doped, Fe-doped, and Fe, N-doped TiO_2_ nanoparticles (0.4 mg/mL) on (**a**) MCF-7 and (**b**) MDA-MB-231 GSSG-to-GSH ratio. Cells treated for 24 h with H_2_0_2_ (1 mM) were used as a positive control of the experiments, untreated used cells as a negative control, and cells treated with visible light without TiO_2_ as an additional negative control. * *p* < 0.05 compared to negative control (untreated) cells. Data represent means ± standard deviation from five independent experiments.

**Table 1 materials-12-02589-t001:** Linear correlation coefficients (R^2^) and photoinduced degradation rate constant (k_app_) of MB for all the TiO_2_ nano-powders.

Doped-TiO_2_	R^2^	k_app_ (min^−1^)
N-TiO_2_	0.98728	22.24 × 10^−3^
Fe,N-TiO_2_	0.90906	4.83 × 10^−3^
Fe-TiO_2_	0.99054	3.5 × 10^−3^
Evonik P25-TiO_2_	0.99074	2.65 × 10^−3^

**Table 2 materials-12-02589-t002:** Linear correlation coefficients (R^2^) and photoinduced degradation rate constant (k_app_) of MB for all the composite particles. A pure IP microgel is included.

Composite Particles	R^2^	k_app_ (min^−1^)
IP microgel/N-TiO_2_	0.98819	9.59 × 10^−3^
IP microgel/Fe,N-TiO_2_	0.9892	2.86 × 10^−3^
IP microgel/Fe-TiO_2_	0.97135	2.3 × 10^−3^
IP microgel/Evonik P25	0.98246	1.16 × 10^−3^
IP microgel	0.82525	2.7 × 10^−4^

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
