# Peer review of "Development of Smart Composites Based on Doped-TiO2 Nanoparticles with Visible Light Anticancer Properties"

_materials, 2019, doi:10.3390/ma12162589_

Round 1

Reviewer 1 Report

This interesting paper deals with the applications of TiO2-based systems as visible light anticancer agents. The manuscript is quite well organized but from my point of view there are some aspects that are not clear.

In particular, the authors should take into consideration the following points:

-General considerations: In all the manuscript are compared different N-doped TiO2 materials, but as described in the experimental the authors used different nitrogen precursors. In particular, they used urea for the N-TiO2 particles and hydroxylamine hydrochloride for the Fe-N-TiO2 particles. Why? The authors should consider this aspect when compare the photocatalytic performance and the anticancer properties.

-The authors carried out the photocatalytic experiments with a lamp that emitted (350-750 nm) in the range of absorption of methylene blue (668 nm). In this contest the photolysis of MB cannot be avoid and more important, the electron transfer from the excited dye to the TiO2, as largely reported in the literature can occur. The high photo-activity under visible light irradiation can be due to this mechanism instead of the band-gap reduction caused by the doping agents. The authors should perform other photocatalytic experiments using another water pollutant which not absorb in the visible range as for example phenol.

Other aspects:

-XRD: Pag. 5 lines 223-224: “Rutile is observed….relevant literature”. The authors should explain why the presence of nitrogen induces the formation of rutile.

Lines 225-226: Why the presence of iron leads to have a broadening of the XRD peaks?

-Figures 2: How the authors have estimated the mean diameter of the particles? It is very difficult from the reported non homogeneous morphology distinguish the nanoparticles.

- Figures and 9: The authors should add the bare TiO2 to can appreciate the red-shift and the band gap variation. Furthermore, I suggest to check again the calculation and the graph-procedure for the determination of the band gap energies. It seems that there is a mid-gap value (or another band-gap) in the N-TiO2 sample ad about 2.18 eV.

-Figure 13: What about the contribution of the adsorption process? (i.e. the variation of concentration in the dark), it can influence the photoactivity.

-Pag. 13 line 388 “which in turns….active sites..” why the formation of iron oxide reduce the number of surface-active sites? some studies report a beneficial effect of the iron oxide (as Fe2O3, band-gap 2.2 eV) in heterojunction with TiO2.

-The authors should report the XPS measurements of the TiO2 embedded in the polymer composite. It would be useful to understand the surface concentration of TiO2 in the polymer matrix to can also rationalized the decrease of photo activity

For my point of view, the manuscript can be published in Materials after major revision according to the above reported comments.

Reviewer 2 Report

In this manuscript, the TiO2 nanoparticles were prepared by utilizing sol-gel technique with different dopants, nitrogen (N-doped), iron (Fe-doped) and nitrogen and iron (Fe, N-doped) and were embedded in an interpenetrating (IP) network micro-gel synthesized by stimuli responsive poly (N-15 Isopropylacrylamide -co-polyacrylicacid) – pNipam-co-PAA forming composite particles. The obtained products are thoroughly characterized by XRD, XPS, Raman, TEM, optical, catalysis, and anticancer peoperties. The manuscript is well organized and contains interesting findings. However, I recommended a major revision of the article from its present form before it can be published in materials. The main concerns are listed below

Comment 1:    The introduction establishes the context of the work, but in the present state it does not provide sufficient justification/motivation for this study. It should be rewritten to expound the research significance of the present work

Comment 2     What is the novelty of the present report?

Comment 3:    How to exclude the sensitization mechanism of MB for the present work?

Comment 4:    The influence of the medium must be studied. Standard deviations of results must be provided (Figures 14)

Comment 5:    The degradation substrate in this paper is methyl blue, which can be applied to the degradation of other organic pollutants?

Comment 6:    The authors should check the loading effect of the catalyst for the degradation of dyes.  

Comment 7:   The author should provide evidences for the environment of recombination of electron–hole pairs, such as, photocurrent, electrochemical impedance spectroscopy, and photoluminescence.

Comment 8:    The explanation of the photodegradation mechanism needs more clarification. Authors are requested to explain the mechanism with neat schematic representation.

Comment 9:    Some evidences should be provided to confirm the proposed photocatalytic mechanism. For example, the active radicals in the photocatalytic process can be investigated by the ESR measurements and the radical-scavenging experiments.

Comment 10:  In the background, the new advances about oxide semiconductor photocatalysts functionalized with an exceptional physicochemical properties for solar-to-chemical energy transformation is a very highly published materials in terms of photocatalysis, authors could consider citing further relevant literature, such as Journal of Alloys and Compounds 735 (2018) 2058, Journal of Materials Science: Materials in Electronics 30 (2019) 10900,

Comment 11:  A bibliographic comparison with other NPs must be done.  

Comment 12:  In the current state, there are more typographical errors and the language should be improved. Therefore, the authors are advised to recheck the whole manuscript for improving the language and structure carefully.  

Reviewer 3 Report

In this study, the authors synthesized TiO2 nanoparticles utilizing sol-gel method with different dopants, nitrogen (N-doped), iron (Fe-doped) and nitrogen and iron (Fe, N-doped). These synthesized doped TiO2 nanoparticles were embedded in an interpenetrating (IP) network microgel synthesized by stimuli-responsive poly (N-16 Isopropylacrylamide -co-polyacrylic acid) – pNipam-co-PAA to forming a nanocomposite. Authors studied morphology and physical properties of the composite and nanoparticles using various characterization techniques. They demonstrated the photocatalytic properties of their samples under visible light irradiation. They found that the N-doped TiO2 nanopowders and composites exhibit the best photocatalytic degradation of the pollutant methylene blue under visible light irradiation. Furthermore, they also studied anticancer behavior by irradiating cultured MCF-7 and MDA-MB-231 breast cancer epithelial using visible light.

Although there have been many reports regarding the doped TiO2 nanoparticles for visible light photocatalysis the authors have done a good job in making a microgel based composite. It provides a light-responsive photocatalysis mechanism for the anticancer properties. I have the following comments for the authors,

1.     The introduction is not well organized, and the problem is not stated clearly. The introduction can be modified so that to give more emphasis on the photocatalytic and anticancer application of the TiO2 rather than the morphology.

2.     Authors should also include the following latest references in the text regarding the photocatalysis and anticancer application of TiO2

a.      R. F. Bonan et al., Mat. Sci. Eng. C, 104, 109876 (2019).

b.     S Kumar et al., RSC Adv., 6, 45120 (2016).

c.      H. Chen et al., ACS Appl Bio. Mater. 1, 1656, (2018).

3.     In some places, the standard rules and style conventions for the use of the SI units are not followed (line 139, 145, 146 to mention a few). At this frequency, the issues are not simply oversights and it is appropriate to ask the authors to have a professional editing service to assist them with language in the manuscript.

4.     Overall the study is correctly designed and technically sound. The work will be of great interest for a wide readership.

Reviewer 4 Report

In this work, Galata et al have explored the potential benefits of nitrogen and/or Fe doping of TiO2 nanoparticles, as well as embedding them in stimuli-responsive microgels. The resulting materials were tested in phototherapy of cancer and photodegradation of organic molecules. This topic of research is a good fit for Materials, and appears to be made with care. The results are interesting, but there are several questions that the authors must address before recommendation for acceptance can be given.

·         Why the focus on visible light for the photo-activation of the TiO2 nanoparticles? It is known that UV and visible light have a low penetration depth into biologic tissues, which generally limits phototherapies to tumors on or just under the skin or on the outer lining of internal organs and cavities;

·         Recycling tests for the photocatalytic studies would be very helpful;

·         Why were the photocatalytic studies performed at pH 6?

·         The photocatalytic mechanism should be identified. Namely, degradation of MB occurs by production of reactive oxygen species (ROS)? What ROS species were produced?

·         Figure 14.a requires error bars;

·         It is said in section 3.4 that cytotoxicity assays were performed with visible light. What was the specific light source, power and wavelength?

·         In what way the TiO2 nanoparticles induce cytotoxicity, when photoactivated? It is by production of reactive oxygen species or by induction of hyperthermia? This must be clarified.

·         The y-axis of Figures 15 and 16 should not be cell number, but cell viability (% of control). In this case, the control should be when the cells are treated with visible light (negative control);

·         Given the authors estimations of the loading efficiency of TiO2 into the microgel, they have compared cytotoxicity assays of non-embedded TiO2 with a concentration of 0.8 mg/mL, with embedded TiO2 with a concentration of ~0.4 mg/mL. The authors need to perform cytotoxicity assays where the embedded and non-embedded TiO2 have the same concentration (either at 0.8 or ~0.4 mg/mL).  

Round 2

Reviewer 1 Report

The authors are well answered  to the all questions. I suggested to improve the quality of the figure 14a because it is very difficult read the legend.

Author Response

The authors have modified Figure 14 as suggested by the Reviewer. Thank you for your valuable suggestion.

Reviewer 2 Report

The authors are responded in a proper manner and it can be accepted in the present form.

Author Response

The authors would like to thank you for your general feedback on the manuscript.

Reviewer 4 Report

While the authors have indeed tried to answer some of my previous questions, there are still some questions that need to be addressed. These can be found on the attached file.

Round 3

Reviewer 4 Report

The manuscript can now be accepted for publication.